# TextCAM: Explaining Class Activation Map with Text

## Abstract

Deep neural networks (DNNs) have achieved remarkable success across domains but remain difficult to interpret, limiting their trustworthiness in high-stakes applications. This paper focuses on deep vision models, for which a dominant line of explainability methods are Class Activation Mapping (CAM) and its variants working by highlighting spatial regions that drive predictions. We figure out that CAM provides little semantic insight into what attributes underlie these activations. To address this limitation, we propose TextCAM, a novel explanation framework that enriches CAM with natural languages. TextCAM combines the precise spatial localization of CAM with the semantic alignment of vision–language models (VLMs). Specifically, we derive channel-level semantic representations using CLIP embeddings and linear discriminant analysis, and aggregate them with CAM weights to produce textual descriptions of salient visual evidence. This yields explanations that jointly specify where the model attends and what visual attributes likely support its decision. We further extend TextCAM to split feature channels into semantically coherent groups, enabling more fine-grained saliency groups with text explanations. Experiments on ImageNet, CLEVR, and CUB demonstrate that TextCAM produces faithful and interpretable rationales that improve human understanding, detect spurious correlations, and preserve model fidelity.

## 1 Introduction

Deep Neural Networks (DNNs) (Krizhevsky et al., 2012; Simonyan & Zisserman, 2015; Vaswani et al., 2017) are widely regarded as black-box models because it is difficult to interpret how they extract features and generate predictions. This opacity poses serious challenges in high-stakes domains such as healthcare, finance, and autonomous systems. To mitigate these concerns, the field of Explainable AI (XAI) has emerged with the goal of making model behavior more transparent and interpretable to humans (Adadi & Berrada, 2018; Minh et al., 2022). XAI techniques enable users and stakeholders to better understand the reasoning behind model outputs and build greater trust in AI-driven decisions (Tjoa & Guan, 2020).

Among the XAI approaches, saliency map (Simonyan et al., 2014; Springenberg et al., 2015) is a common technique used to explain deep vision models. A mainstream of techniques are Class Activation Mapping (CAM) and its variants, which highlight spatial regions that contribute most to a prediction, offering an intuitive visualization of where the model focuses (Zhou et al., 2016; Wang et al., 2020; Selvaraju et al., 2017; Chattopadhay et al., 2018a; Jiang et al., 2021). However, the information they provide remains limited. Heatmaps indicate evidence locations but not the nature of the evidence. For example, in fine-grained bird recognition, a CAM may highlight the beak, yet it is unclear whether the model relies on the beak's shape, color, or texture. This under-specification hinders interpretability for human users, restricts diagnostic value for practitioners, and limits the ability to detect spurious correlations (e.g., reliance on background cues) (Beery et al., 2018).

At the same time, the rise of vision–language models (VLMs) such as CLIP has demonstrated the power of large-scale image–text alignment. By embedding both modalities into a shared representation space, VLMs enable zero-shot recognition and natural-language querying (Radford et al., 2021; Jia et al., 2021; Zhai et al., 2021). Importantly, they provide a more human-interpretable modality for reasoning about visual content: attributes and concepts can be directly expressed in

text. Recent work has therefore begun to explore the explainability of VLMs themselves, analyzing their architectures or probing their training dynamics (Li et al., 2022; Mu et al., 2022). Yet despite these advances, the potential of natural language as an explanation tool for pure vision models remains largely underexplored.This is a critical gap, as convolutional and transformer-based vision models continue to dominate real-world deployments due to their efficiency, maturity, and regulatory acceptance. The challenge is that, although VLMs achieve powerful image-text alignment, this multi-modal capacity doesn't naturally apply to the decision process of black-box vision models.

In this work, we aim to bridge this gap by introducing TextCAM, a novel method that enriches CAM-based explanations with natural-language rationales. Our key insight is that CAMs provide high-precision spatial cues about *where* evidence lies, while VLMs provide high-recall semantic spaces describing *what* attributes might be present. By combining these two signals, TextCAM produces textual explanations that specify not only the region of focus but also the underlying visual attributes that most plausibly drive the model's decision. This paper makes the following contributions: (1) We introduce TextCAM, a framework that enriches Class Activation Maps with natural-language rationales by bridging spatial localization from CAM with semantic alignment from vision–language models. (2) We develop a method to compute channel-level semantic representations via CLIP embeddings and linear discriminant analysis, enabling faithful mapping from visual activations to textual attributes. (3) We design a sparse text selection and grouping strategy that produces concise, diverse explanations and partitions feature channels into semantically coherent groups, yielding fine-grained text-annotated saliency maps. (4) We conduct extensive experiments on ImageNet, CLEVR, CUB-200, and DomainNet. TextCAM demonstrates decent interpretability and faithfulness. Our experiments also show potential applications of feature engineering with TextCAM.

## 2 RELATED WORK

### 2.1 MACHINE LEARNING INTERPRETABILITY

Explainable Artificial Intelligence (XAI) aims to make complex machine learning models more transparent, addressing the "black-box" nature of many advanced AI systems (Nauta et al., 2023). Its primary objective is to generate explanations that are understandable to humans, which is essential for fostering trust, ensuring accountability, and promoting ethical AI deployment (Kusner et al., 2017). Existing approaches to XAI include local optimization methods (Lundberg & Lee, 2017), occlusion-based techniques (Fong et al., 2019; Petsiuk et al., 2018), gradient-based strategies (Baehrens et al., 2010; Rebuffi et al., 2020), and class activation map (CAM)-based approaches (Muhammad & Yeasin, 2020; Ramaswamy et al., 2020; Selvaraju et al., 2017).

### 2.2 EXPLAINING COMPUTER VISION MODELS

In visual explanation research, saliency mapping is widely employed in image classification, as it identifies the image regions most critical to a model's decision while leaving the underlying model unchanged (Dabkowski & Gal, 2017; Kapishnikov et al., 2019). Among existing approaches, Grad-CAM (Selvaraju et al., 2017) has become one of the most popular, producing heatmaps that emphasize regions most responsible for a model's predictions. A number of extensions, such as Grad-CAM++ (Chattopadhay et al., 2018b), Score-CAM (Wang et al., 2020), Ablation-CAM (Ramaswamy et al., 2020), XGrad-CAM (Fu et al., 2020), Eigen-CAM (Muhammad & Yeasin, 2020), Layer-CAM (Jiang et al., 2021), Smooth Grad-CAM++ (Omeiza et al., 2019), DiffCAM (Li et al., 2025) and FinerCAM (Zhang et al., 2025) have been proposed to enhance interpretability, applicability or produce more discriminative saliency maps. Beyond CAM-style techniques, other saliency methods like Gradient × Input (Simonyan, 2013) and Integrated Gradients (Sundararajan et al., 2017) take a different route: instead of relying on deep features, they directly compute contributions from the input space to identify critical evidence influencing the model's decisions. Despite their usefulness, these saliency mapping methods suffer a big limitation that they only indicate important positions in the image, without explaining what visual patterns are extracted in the position. Another research direction, network dissection (Bau et al., 2017; 2020; Kim et al., 2018), aims to explain individual neurons with semantic concepts by utilizing a pre-defined concepts or a dataset of annotated semantic maps.

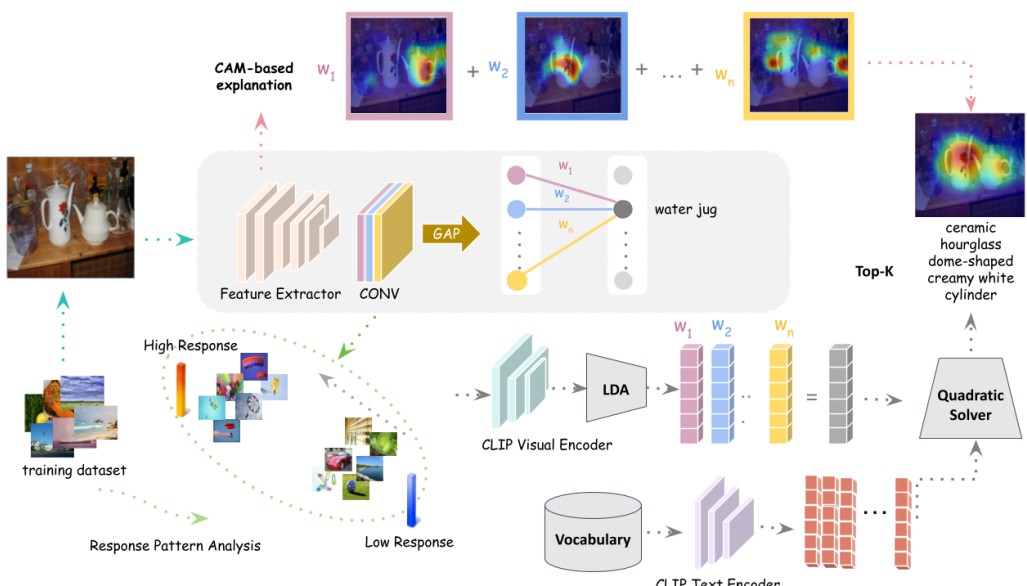

Figure 1: Overview of TextCAM. *Left Bottom*: Per-channel response pattern analysis with positive/negative samples. Per-channel representation is calculated by LDA in the image-text joint space of CLIP. *Right Bottom*: Calculating overall semantic representation by weights from CAM and selecting a diverse set of text explanations using sparse optimization. *Right Top*: Explaining saliency maps with top-K text explanations.

## 2.3 VISION-LANGUAGE MODELS

Contrastive Language-Image Pretraining (CLIP) (Radford et al., 2021) has driven a major shift in zero-shot and few-shot image classification. Subsequent works further turn images into text-like tokens so LLMs can parse visual inputs (Ye et al., 2023; Zhu et al., 2023). Instruction-tuned LVLMs (e.g., InstructBLIP, LLaVA) support curated human instructions for stronger visual reasoning (Dai et al., 2023; Liu et al., 2023; Lin et al., 2024).

Explainable Artificial Intelligence (XAI) for Vision–Language Models (VLMs) has emerged as a crucial research direction to address the "black-box" nature of multimodal models such as CLIP. Class Activation Values (CAV) (Chen et al., 2025) provides fine-grained interpretations for CLIP by combining class-specific gradients and multi-scale activations in the image encoder with relevance attribution in the text encoder. Beyond CAV, Goh et al. (2021) identified multimodal neurons encoding semantic concepts across modalities, Materzyńska et al. (2022) studied the entanglement of word images and natural images, and Gandelsman et al. (2023) leveraged CLIP's embedding space and architectural decomposition to offer more systematic interpretations.

The above methods are specifically designed for interpreting vision-language models (VLMs) such as CLIP, whereas our approach focuses on explaining vision models, enabling multimodal interpretability in purely visual tasks.

## 3 TEXTCAM METHOD

TextCAM is a multimodal explanation method built upon Class Activation Maps (CAM) (Zhou et al., 2016). While CAM provides only visual saliency maps as post-hoc explanations of model's predictions, TextCAM complements the saliency map with text-based descriptions, thereby integrating both visual and semantic information. The overall framework of TextCAM is illustrated in Figure 1.

## 3.1 BACKGROUND OF CAM

For simplicity, we assume the model to be explained consists of a deep feature extractor followed by a shallow classification head, which is a common design in modern DNNs. Let $\mathbf{x}$ denote the target input, and let $d$ be the number of feature channels produced by the feature extractor. The forward pass yields activation maps $\mathbf{A}_j(\mathbf{x})$ for each channel $j \in \{1, 2, \ldots, d\}$. The objective of CAM-based approaches is to generate a saliency map $\mathbf{V}_c(\mathbf{x})$ that highlights the regions which contribute most to the prediction for class $c$. This is achieved by linearly combining the activation maps with a feature importance vector $\mathbf{w}^c \in \mathbb{R}^d$:

$$\mathbf{V}_c(\mathbf{x}) = \sum_{j=1}^{d} \mathbf{w}_j^c \, \mathbf{A}_j(\mathbf{x}). \tag{1}$$

In the original CAM method (Zhou et al., 2016), the importance weights $\mathbf{w}_j^c$ are directly taken from the parameters of the linear classification head: for class $c$, $\mathbf{w}_j^c$ corresponds to the $j$-th entry of the fully connected layer's weight vector $\mathbf{W}_c$. Many subsequent methods adopt this general framework but differ in how they compute $\mathbf{w}^c = \mathcal{CAM}(\mathbf{x}, c)$. Our approach is agnostic to the specific CAM variant and feasible to any of the CAM methods discussed in Related Work. We reuse the computed importance weights $\mathbf{w}$ from visual CAM, and then generate semantic explanations instead of saliency maps based on $\mathbf{w}$.

## 3.2 TEXTCAM DESIGN

**Analyze Per-channel Response Patterns.** To obtain the semantic representation $\mathbf{s}_j$ for each channel $j$, we first propagate the training dataset through the network to extract the $d$-dimensional feature maps from the target layer to be explained. Following common practices, we use the last convolutional layer by default. For channel $j$, we compute an activation score by applying global average pooling (GAP) over the corresponding feature map. Based on these scores, we select the $M$ samples with the highest responses as positive examples and the $M$ samples with the lowest responses as negative examples. The resulting $2M$ samples are assigned binary labels (positive vs. negative) according to their activation scores. Note that class labels from the dataset are not needed in this process. These labeled samples thus provide a data-driven characterization of channel $j$.

**Generate Semantic Representation.** The next step is to represent each channel by a latent vector. We adopt the idea that the most expressive representation should be the latent visual feature that best distinguishes high- and low-response examples. To achieve this, we apply Linear Discriminant Analysis (LDA) in the CLIP embedding space, since LDA solves the projection direction that maximizes the inter-class (i.e. positive vs negative) variance and minimizes the intra-class variance. Specifically, we first encode the selected samples using the image encoder of CLIP to obtain their embeddings. Then we calculate the optimal projection vector $\mathbf{p}_j$ for each channel $j$ with LDA based on these embeddings with the assigned binary labels. To further align visual CAM whose activation levels contribute to spatial importance in saliency maps, we also consider per-channel activations $a_j = \text{GAP}(\mathbf{A}_j)$ when aggregating the semantic representation. Specifically, we have $\mathbf{s}_j = a_j \mathbf{p}_j$. The semantic representation is then calculated as

$$\mathbf{T}_c(\mathbf{x}) = \sum_{j=1}^{d} \mathbf{w}_j^c \mathbf{s}_j(\mathbf{x}). \tag{2}$$

**Sparse Text Selection.** To represent $\mathbf{T}_c(\mathbf{x})$ with interpretable natural languages, we first prepare a vocabulary where each item is a word or phrase indicating a visual pattern or concept (more details in Appendix A). The text embedding for each item is pre-calculated using the text encoder of CLIP, resulting in the embedding matrix $\mathbf{E} \in \mathbb{R}^{d \times N}$, where $N$ is the total number of candidate text descriptions and $d$ is the feature dimension. Our goal is to select a few items which best align the target representation $\mathbf{T}_c(\mathbf{x})$. Regarding interpretability, we also favor the selected descriptions to be more diverse. This interpretation task can be formulated as a sparse approximation problem with correlation-aware regularizations. Denoting $\omega \in \mathbb{R}^N$, the goal is to solve

$$\omega^* = \arg\min_{\omega \geq 0} \frac{1}{2}||\mathbf{T}_c - \mathbf{E}\omega||_2^2 + \alpha||\omega||_1 + \beta\omega^T\mathbf{G}_{\text{off}}\omega, \tag{3}$$

where $||\omega||_1$ encourages sparsity, and $\mathbf{G}_{\text{off}} = (\mathbf{1} - \mathbf{I}) \odot \mathbf{E}^T\mathbf{E}$ is the covariance matrix with zeros on the diagonal, which penalizes correlations (i.e., text descriptions with very similar meanings) among the selected embeddings. This quadratic problem can be efficiently solved by Alternating Direction Method of Multipliers (ADMM) (Boyd et al., 2011). We get the words or phrases corresponding to the solved top-$K$ non-zero items as the final text explanation for the saliency map $\mathbf{V}_c$.

### 3.3 TextCAM for Saliency Groups

We can extend TextCAM to support more informative visual and text explanations by splitting the activation maps into several groups based on the selected top-$K$ text descriptions $t_1, t_2, ..., t_K$. Formally, our task is to perform partition of the channel indices $\{1, \ldots, d\}$ into $K$ non-empty disjoint groups $G_1, \ldots, G_K$. The assignments are represented by a set of integers $\{g_j\}_{j=1}^d$, with $g_j = k$ indicating channel $j \in G_k$.

Denote the selected text embeddings by $\{\mathbf{e}_j\}_{j=1}^K$, and the $j$-th weighted text embedding as $\bar{\mathbf{s}}_j = \mathbf{w}_j^c\mathbf{s}_j(\mathbf{x})$. Since our goal is to find the channel group that best describe each solved semantic $t_j$, we use $\{\mathbf{e}_j\}_{j=1}^K$ as $K$ fixed target centers, and solve $\{g_j\}_{j=1}^d$ by minimizing the total squared deviation of each group mean from its embedding center as

$$\{g_j^*\}_{j=1}^d = \arg\min_{\substack{g_j \in \{1, \ldots, K\} \\ G_k \neq \varnothing}} J(\mathbf{g}) = \sum_{k=1}^K n_k \|\mu_k - \mathbf{e}_k\|_2^2, \tag{4}$$

where $n_k = |G_k|$ is the weight for group $k$, and $\mu_k = \frac{1}{n_k}\sum_{j \in G_k}\bar{\mathbf{s}}_j$ is the empirical mean of group $k$. We use a greedy relocation algorithm (described in Appendix B) adapted from Kanungo et al. (2002) to solve this NP-hard problem.

With the partition result, we can now generate a separate saliency map for each selected text description. Specifically, for the $k$-th group, the saliency map (corresponding to text $t_k$) is:

$$\mathbf{V}_c^k(\mathbf{x}) = \sum_{j \in G_k} \mathbf{w}_j^c\mathbf{A}_j(\mathbf{x}). \tag{5}$$

By this extension, TextCAM not only explains saliency maps with more interpretable text descriptions, but also provides more fine-grained form of explanations by text-annotated saliency groups.

## 4 Experiments

We evaluate TextCAM across datasets, architectures, and multiple CAM families. Our study focuses on three questions: (1) whether TextCAM can deliver word-level explanations that are consistent with the CAM spatial evidence while operating in the CLIP joint text–image space; (2) whether saliency grouping with TextCAM can stably aggregate activated channels by *text concepts* and expose relatively independent regions through concept-driven separation/inspection; (3) how broadly the approach adapts across backbones, layers, and different sources of channel weights $\mathbf{w}^c$.

### 4.1 Datasets and Models

We evaluate our method on ImageNet-1k (Deng et al., 2009), CUB-200-2011 (Wah et al., 2011), and a balanced CLEVR subset (Johnson et al., 2017). CLIP ViT-B/32 provides the cross-modal space; explanations are computed on standard CNN/Transformer backbones (ResNet-50 (He et al., 2016), Swin-Transformer (Liu et al., 2021)) at the last stage; all runs use public checkpoints and evaluation mode.

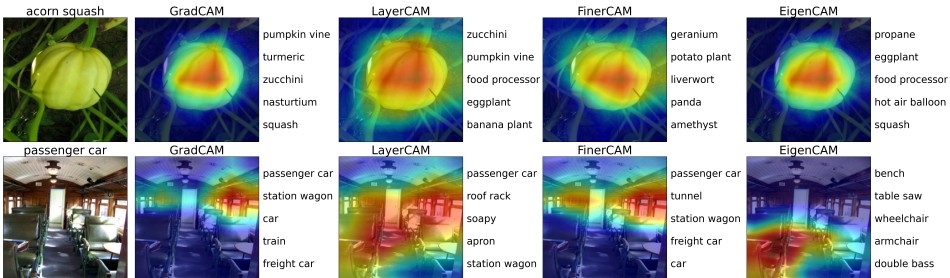

Figure 2: **ImageNet results of TextCAM.** Each column displays, respectively, the original image and the TextCAM results using Grad-CAM, Layer-CAM, Finer-CAM, and Eigen-CAM.

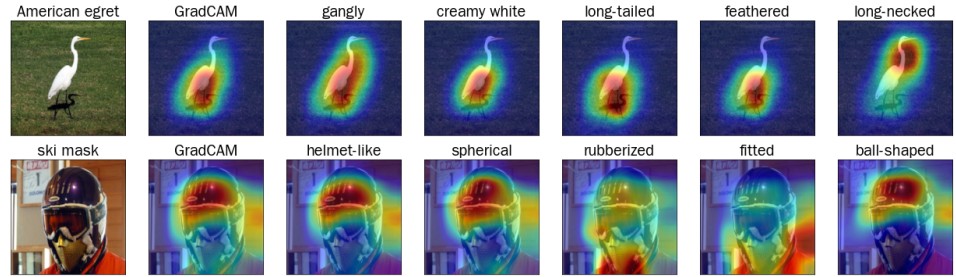

Figure 3: **ImageNet results of TextCAM with saliency groups.** Each column displays, respectively, the original image, Grad-CAM result, and grouped saliency maps along with their corresponding text explanations from the top-5 TextCAM results.

### 4.2 QUANTITATIVE EVALUATION

**ImageNet and CUB-200 results.** We evaluate TextCAM on the ImageNet validation set using a broad, domain-agnostic vocabulary constructed with ChatGPT (Achiam et al., 2023) from five concept families: color descriptors, texture descriptors, shape descriptors, abstract concepts, and entities (animals, scenes, plants, man-made objects, and their parts). We adopt ResNet as the backbone. For the channel weights $\mathbf{w}^c$, we plug in four representative CAM families-Grad-CAM, Layer-CAM, Finer-CAM, and Eigen-CAM-without modifying any backbone parameters. As summarized in Fig. 2, TextCAM yields qualitatively consistent and semantically plausible phrases across all $\mathbf{w}$ sources. Because our method derives $\mathbf{w}$ directly from the chosen CAM, the resulting textual explanations remain faithful to the corresponding spatial evidence-even in cases where the CAM emphasizes non-subject regions. Among alternatives, Grad-CAM provides the most stable weighting; consequently, TextCAM yields stable, intuitive outputs. Further results on CUB-200 are included in 9 in Appendix.

**Saliency grouping result.** We present qualitative examples of TextCAM-based saliency groups in Fig. 3. The two examples demonstrate the typical effects of TextCAM for generating separate saliency map groups for the individual text explanations. Most of the saliency groups highlight the reasonable spatial region corresponding to the text explanation. For example, when explaining the image of American egret, the visual attributes gangly, feathered and long-necked are successfully tied with saliency groups focusing on different regions. More qualitative examples are presented in Fig. 8 in Appendix.

### 4.3 FAITHFULNESS AND SPECIFICITY ANALYSES

#### 4.3.1 CONCEPT ALIGNMENT UNDER CONTROLLED SHIFTS

**CLEVR benchmark.** We test whether TextCAM aligns with ground-truth semantics on CLEVR, where attributes are factorized and controlled shifts are available. We render a balanced corpus over four color–shape combinations (red/blue × cube/ball) and hold out object layouts across splits

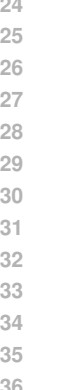

Figure 4: **CLEVR qualitative results.** Each row shows the input image, TEXTCAM for the *shape* head ($M_a$), and TEXTCAM for the *color* head ($M_b$). Top row: *blue cube*; bottom row: *red ball*. In both cases, heatmaps concentrate on the target object while remaining insensitive to the *yellow cylinder* distractor, indicating that the retrieved concepts (shape versus color) are supported by spatially localized evidence rather than incidental context.

to discourage positional shortcuts. A frozen image backbone feeds two linear heads: a *shape head* $M_a$ (cube vs. ball) and a *color head* $M_b$ (red vs. blue). At test time only, we optionally add 1–3 non-overlapping `yellow cylinders` as salient but label-irrelevant distractors. In all CLEVR analyses there are **two** models: $M_a$ (shape) and $M_b$ (color). For a given image, explanations are judged against the attribute relevant to the queried head (shape for $M_a$, color for $M_b$). Concretely, if an image shows a *red ball*, then the correct textual concept for $M_a$ is `ball` and for $M_b$ is `red`. We use this head-specific notion of correctness throughout. To make the contrast intuitive, we also include a two-image demonstration (one `blue cube`, one `red ball`) in which Grad-CAM heatmaps from $M_a$ and $M_b$ are nearly identical (both focus on the object), yet TextCAM produces distinct *words* per head on the very same pixels (cube vs. `blue`, `ball` vs. `red`); this illustrates that textual explanations communicate *what property* the model uses, rather than only *where* it looks.

**Models and TEXTCAM inference.** Given an image $x$ and a trained head $M \in \{M_a, M_b\}$, TEXTCAM extracts a discriminative signal (Eq. (2)) and forms a concept vector $\mathbf{T}_c(x) \in \mathbb{R}^d$. We fix a bank $\mathcal{C} = \{\text{red}, \text{blue}, \text{yellow}, \text{cube}, \text{ball}, \text{cylinder}\}$, and pre-compute CLIP text embeddings $\{\mathbf{e}_c\}_{c \in \mathcal{C}}$. For rigor, we $\ell_2$-normalize both vectors before computing cosine similarity, i.e., $\tilde{\mathbf{T}}_c(x) = \mathbf{T}_c(x)/\|\mathbf{T}_c(x)\|_2$ and $\tilde{\mathbf{e}}_c = \mathbf{e}_c/\|\mathbf{e}_c\|_2$, and score

$$s_M(x, c) = \cos\big(\tilde{\mathbf{T}}_c(x), \tilde{\mathbf{e}}_c\big), \qquad \hat{c}_M(x) = \arg\max_{c \in \mathcal{C}} s_M(x, c).$$

We visualize Top-$K$ concepts ($K$=5) and report Top-1 accuracy over 100 held-out images per head. To evaluate textual correctness for each head we define

$$\text{Acc}^{\text{TXT}}(M) = \frac{1}{N} \sum_{i=1}^{N} \mathbf{1}\big\{\hat{c}_M(x_i) = y_M^\star(x_i)\big\},$$

where $y_M^\star(x)$ is the relevant attribute for the head (*shape* for $M_a$, *color* for $M_b$). Both heads reach $\text{Acc}^{\text{TXT}}$=100% on the CLEVR subset, including distractor scenes, showing that TEXTCAM retrieves the intended concepts on a per-image basis. Under this protocol, the textual Top-1 accuracy for both heads is perfect:

$$\text{Acc}^{\text{TXT}}(M_a) = 100\%, \qquad \text{Acc}^{\text{TXT}}(M_b) = 100\%.$$

Thus, for every one of the 100 held-out images per head, including distractor scenes, TEXTCAM retrieves the correct *shape* term for the shape head and the correct *color* term for the color head.

### 4.3.2 APPLICATION: TEXTCAM-GUIDED DEBIASING FOR SHAPE RECOGNITION

**Setup: biased-shape classifier.** We next study whether TEXTCAM-guided inspection can *improve* out-of-distribution performance for a 3-way *shape* classifier over $\{\text{cube}, \text{ball}, \text{cylinder}\}$ when

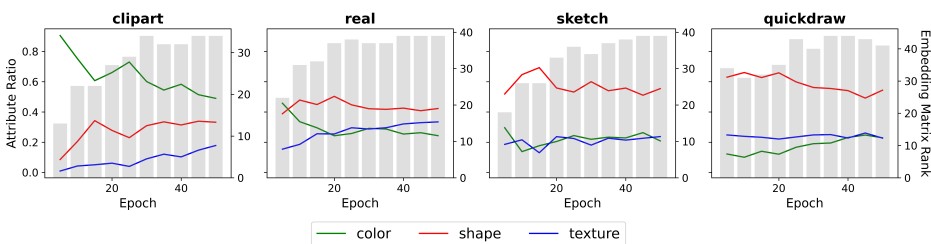

Figure 5: TextCAM result statistics for ResNet50 models trained on clipart, real, sketch and quick-draw domains from the DomainNet dataset. Colorful curves represent the ratio of each attribute type appearing in the top-1 TextCAM results. Gray bars represent the approximate rank of embedding matrix formed by the top-1 TextCAM text embeddings from 1000 test examples.

the training data exhibit a spurious color–shape correlation (cube: $90\%$ `blue`; ball: $90\%$ `red`; cylinder: $90\%$ `yellow`). The model is a frozen ResNet-50 with global average pooling producing $\mathbf{z}(x) \in \mathbb{R}^d$; let $d = 2048$. A frozen linear head $W_{\text{shape}} \in \mathbb{R}^{3 \times d}$ maps features to shape logits. We evaluate the model on a color-balanced test set so that the results reflect shape recognition ability rather than reliance on color.

**Color probe and channel selection.** To reveal color-bearing structure in $\mathbf{z}$, we fit a bias-free linear *color probe* $W_{\text{color}} \in \mathbb{R}^{3 \times d}$ on frozen features using labels parsed from filenames (`red`/`blue`/`yellow`). Because the probe is linear and bias-free, each row $W_{\text{color}}[c, \cdot]$ has a CAM-style interpretation over channels. We score channels by $|W_{\text{color}}[c, j]|$, select Top-$K$ per color, and take the union

$$\mathcal{S} = \bigcup_{c \in \{\text{red}, \text{blue}, \text{yellow}\}} \text{TopK}(|W_{\text{color}}[c, \cdot]|, K),$$

yielding a representation-space of *color-dominant mask*. Intuitively, $\mathcal{S}$ spans directions that are linearly sufficient for color discrimination under the biased training distribution; we hypothesize that the shape head partly projects onto this subspace, creating a color→shape shortcut.

**Inference-time intervention and sensitivity.** We perform a training-free edit at test time by suppressing color-dominant coordinates before the frozen shape head:

$$z_j^{(\text{abl})} = \begin{cases} 0, & j \in \mathcal{S}, \\ z_j, & \text{otherwise}, \end{cases} \qquad \hat{\mathbf{y}} = W_{\text{shape}} \, \mathbf{z}^{(\text{abl})}.$$

Here $\mathcal{S}$ is mined *offline* from the training split using a bias-free linear color probe on the backbone's penultimate features. We take the Top-$K{=}64$ dimensions per color (`red`/`blue`/`yellow`) by $|w|$ from the probe's CAM-style weights and union them to form $\mathcal{S}$, yielding $|\mathcal{S}|{=}175$ ($8.54\%$ of channels). During intervention and evaluation all network weights remain frozen; features are extracted under `no_grad`, and the edit is applied only at inference.

**Biased-shape intervention via ablation of the color subspace:** Using the predefined loaders, Top-1 on the **validation** split ($N{=}300$, same $90\%$ color–shape bias as training; held-out layouts) improves from $\mathbf{0.8767}$ to $\mathbf{0.9533}$ ($+\mathbf{7.66}\,\text{pp}, +\mathbf{8.74}\%$). On the **test** split ($N{=}300$, color-balanced with one-third of each color within shape; held-out layouts), Top-1 rises from $\mathbf{0.7567}$ to $\mathbf{0.8433}$ ($+\mathbf{8.66}\,\text{pp}, +\mathbf{11.44}\%$). The intervention reduces shortcut reliance: after ablation, TEXTCAM Top-$K$ concept frequencies shift from color to shape, while Grad-CAM maps remain spatially stable, indicating that the gains arise from reweighting *what* evidence is used rather than changing *where* the model attends.

### 4.3.3 SENSITIVITY

**DomainNet benchmark.** The DomainNet (Peng et al., 2019) dataset is one of the largest and most diverse benchmarks for domain adaptation, and it is particularly well-suited for validating how DNNs extract and rely on different types of visual attributes. With about 0.6 million images across 6 domains and 345 shared object categories, the dataset naturally highlights distinct attribute biases.

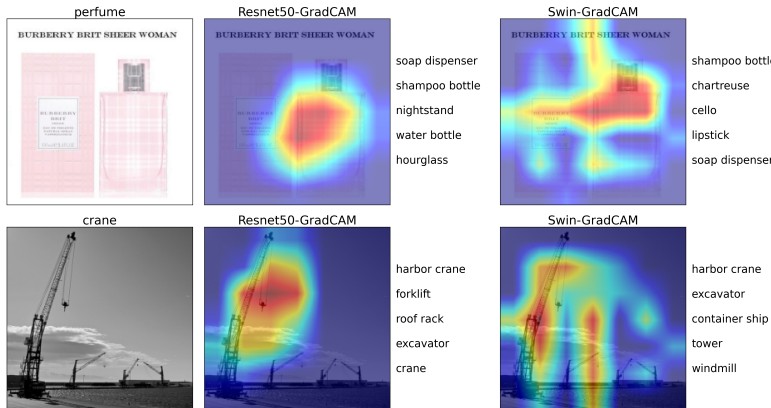

Figure 6: **TextCAM on a Swin Transformer backbone.** TextCAM maintains consistent explanations, though some transformer models yield less stable **w** from Grad-CAM-like procedures.

For instance, the *sketch* and *quickdraw* domains lack rich color information, making shape features far more critical for recognition compared with *real* and *clipart* domains. This clear variation in attribute dependence provides a natural ground truth for evaluating whether the attribute statistics derived from TextCAM align with common sense expectations about domain-specific visual cues. Some image samples from the dataset are shown in Appendix Fig. 10.

**Evaluation Strategy.** We use four domains *sketch*, *quickdraw*, *real* and *clipart* in our experiment due to their distinguishable visual characteristics. For each domain, we randomly select 100 classes and train a ResNet-50 model from scratch for 50 epochs. We save the checkpoint for every 5 epochs. TextCAM analysis is computed on the checkpoint trajectory. To ensure accurate faithfulness evaluation of TextCAM, we use an abbreviated vocabulary by keeping only 590 descriptive items, categorized into three atribute types: color, shape and texture. We use 1000 test examples as the test set. For each test image, we record its top-1 TextCAM explanation.

**Result Analysis.** We first analyze the correlation between TextCAM explanations and the known visual patterns of the selected domains. We type each TextCAM explanation and compute the occurrence ratio per attribute. As shown in Fig. 5, we find that the attribute statistics align well with the dataset characteristics. For example, TextCAM results show that sketch and quickdraw mostly extract shape attributes, whereas clipart rely relatively more on color patterns. The TextCAM results also match common learning dynamics. For each checkpoint, we first prepare the embedding matrix formed by the top-1 explanation for each test image. Then we calculate the energy-based approximate rank of the matrix by keeping 95% of the leading singular values. It can be observed that, as training proceeds, models gradually learn more diverse semantic types, i.e., with increased rank of the embedding matrix and more uniform distribution among different attribute types.

### 4.4 EXTENSION TO VISION TRANSFORMER

We extend TextCAM to transformer-based networks by treating all non-[CLS] tokens as the spatial grid of size $H \times W$. As shown in Fig. 6, the method produces consistent explanations under transformer architectures as well. For certain transformer variants, CAM-derived channel weights **w** (e.g., from Grad-CAM-style formulations) can be less stable due to token mixing and attention reweighting; nevertheless, our text retrieval remains coherent given the provided **w**.

## 5 CONCLUSION

We presented TextCAM, a training-free method that couples CAM spatial evidence with CLIP semantics to deliver faithful, concise explanations and concept-grouped saliency. The approach is architecture-agnostic and complements existing CAM variants without altering model weights. Current limitations include sensitivity of CAM-style weights in some transformer backbones.

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

## A  VOCABULARY PREPARATION

The vocabulary in this work is designed with the guiding principle that it should comprehensively cover fundamental visual attributes and concepts, so that the explanations can capture a wide range of patterns a model might rely on.

By default, we curate an adjective–noun list spanning colors, materials, textures, shapes, objects, and object parts, and further extend it for ImageNet using ChatGPT to ensure coverage of broad semantic families, including color descriptors, texture descriptors, shape descriptors, abstract concepts, and entities such as animals, plants, scenes, and man-made objects. We use the following template to prompt ChatGPT for vocabulary construction.

> **Prompt to ChatGPT-4o**
>
> Generate an as-comprehensive-as-possible set of atomic English descriptors—single words or fixed short phrases in standard usage—that can be used to describe things, including: color descriptors, texture descriptors, shape descriptors, abstract concepts, and entities (various animals, landforms, plants, man-made artifacts, and their component parts).

For domain-specific datasets and simulation tasks, we construct the corresponding vocabulary case-by-case. For CUB, the released fine-grained attribute annotations (e.g., bill shape, throat color) are paraphrased into short natural phrases and incorporated. For CLEVR, the vocabulary is constructed around its controlled attributes (e.g., cube, ball, red, blue) with extra distractor descriptors (e.g., yellow cylinder) included for robustness. For DomainNet, only visual attributes describing color, shape and texture are used as candidates for quantitative evaluation.

All entries are embedded into CLIP's text space and cached, enabling efficient matching against the semantic representation of feature channels. Please refer to the experiment section for more details.

## B  CHANNEL ASSIGNMENT ALGORITHM

Our algorithm is mostly motivated by classical local search algorithms for clustering (Kanungo et al., 2002). We make slight modification to adapt the original algorithm to our problem with fixed group centers. The basic idea of the greedy relocation algorithm for channel assignment is that, in each iteration, we reassign a single point (i.e., the assignment of a channel) to a different cluster (i.e., another semantic group) if doing so reduces the objective defined in Eq. (4). Here are the detailed implementation.

We first assign each channel $j$ (with the corresponding semantic representation $\bar{s}_j$) to the nearest group center $e_k$ as initialization. For each group $G_k$, we then compute its point count $n_k$, sum of embedding $S_k = \sum_{i:g_i=k} e_i$ and mean of embedding $\mu_k = \frac{S_k}{n_k}$.

Then we perform iterative single-point move attempts. Specifically, for each point $s$ currently in cluster $a$, consider moving it to another cluster $b \neq a$. The operation is determined by the objective change, which can be computed without recomputing all means.

This is how to efficiently compute the objective change. We first compute the current objective contributions of group $a$ and $b$ separately as $J_a = n_a \|\mu_a - e_a\|^2$, $J_b = n_b \|\mu_b - e_b\|^2$. If we move $s$ from $a$ to $b$, the objective change consists of the following two parts.

(1) For group $a$, we have: $\mu'_a = \frac{S_a - s}{n_a - 1}$, $\quad J'_a = (n_a - 1)\|\mu'_a - e_a\|^2$ $\quad$ (skip if $n_a = 1$).

(2) For group $b$, we have: $\mu'_b = \frac{S_b + s}{n_b + 1}$, $\quad J'_b = (n_b + 1)\|\mu'_b - e_b\|^2$.

Therefore, the total objective change can be computed as $\Delta = (J'_a + J'_b) - (J_a + J_b)$. If $\Delta < 0$, we perform the move and update $S_a, S_b, n_a, n_b, \mu_a, \mu_b$. An ideal implementation is to repeat the move attempts until no improving move exists. In practice, we set the maximum sweep time as 5,000 for efficiency.

## C    IMPLEMENTATION DETAILS ON LARGE-SCALE DATASETS

**Channel representations (reference data and hooks).** For both ImageNet and CUB we use their respective *test/validation* images as the reference pool to estimate channel semantics. For CLEVR, we use the *training* split without distractors as the reference pool. With the reference pools fixed, for CNNs we hook the forward activations at the target layer, and for transformers we average the spatial tokens (excluding the classification token) to obtain a per-channel activation score compatible with CAM aggregation.

**Positive/negative selection and LDA.** Channel-wise discriminative directions are estimated via LDA in the CLIP image-embedding space. For each channel we select images with the highest and lowest activations as positives/negatives. We considered {50, 100, 150, 200} images per side and found **100** to be a robust default across datasets/backbones. CLIP's image encoder provides the embeddings on which LDA is computed; the resulting per-channel vectors remain in the same space as text embeddings and thus are directly comparable.

**Image-level representation (TextCAM).** Given per-channel vectors, TextCAM forms an image-level representation by a weighted sum using channel weights $\mathbf{w}^c$. We plug in different CAM families as sources of $\mathbf{w}^c$: gradient-based (Grad-CAM, Layer-CAM), forward-based (Score-CAM), and PCA-based (Eigen-CAM), among others. For Layer-CAM we approximate the channel weights with $\mathrm{ReLU}(\nabla)$ at the target layer.

**Text retrieval & saliency group rendering.** All vocabulary entries are encoded by the CLIP text encoder once and cached. Given the TextCAM image vector $T_c(x)$, we retrieve the Top-$K$ (default $K{=}5$) phrases via sparse approximation. For the greedy relocation of saliency grouping, we cap the number of sweeps at 5000. After assignment, each group's sub-CAM is rendered by summing the group's feature maps weighted by their class-specific channel weights; we apply ReLU and linear normalization to $[0, 1]$ for visualization.

## D    QUALITATIVE EXAMPLES FOR TEXTCAM

## E    DOMAINNET EXAMPLES

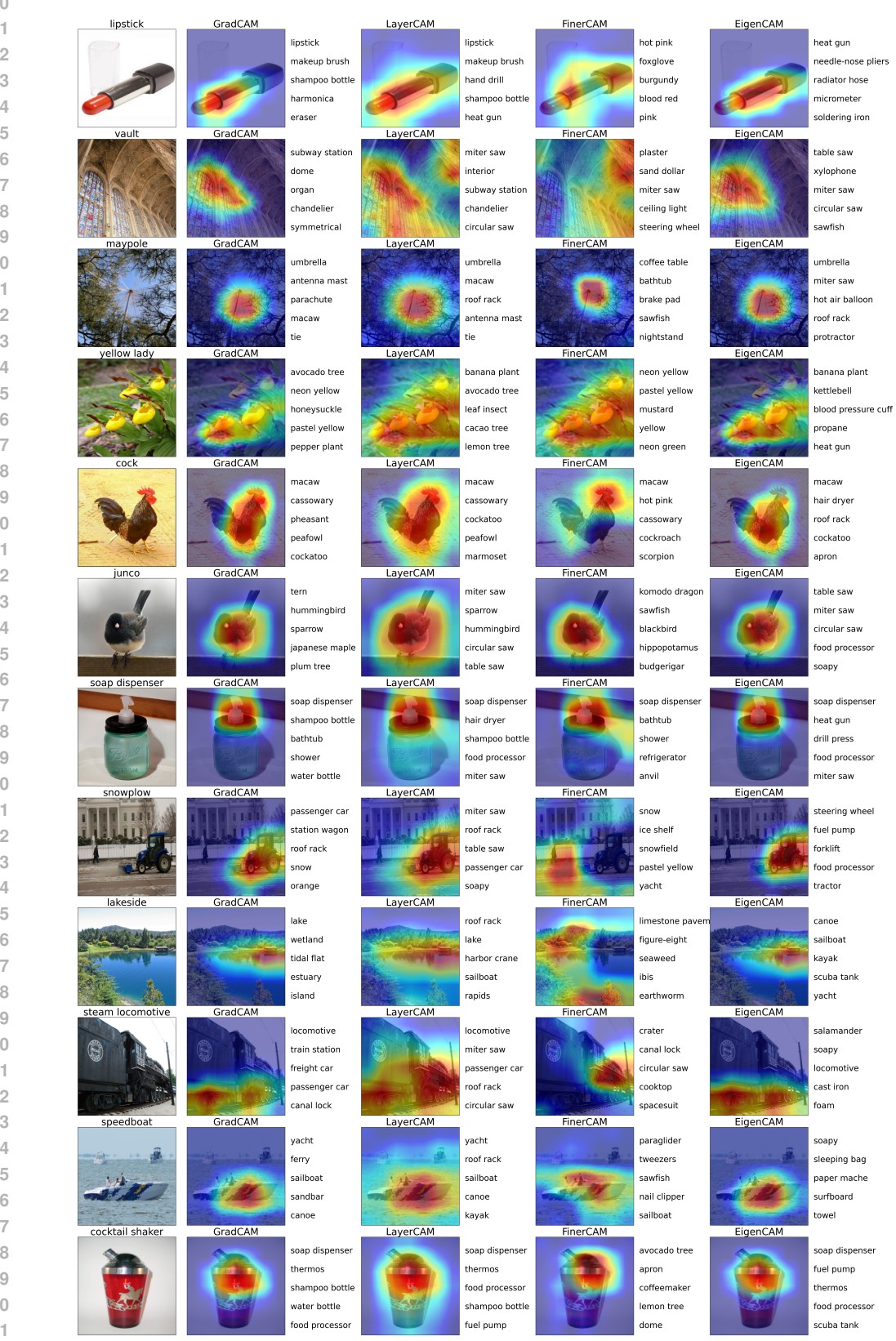

Figure 7: **ImageNet results of TextCAM**

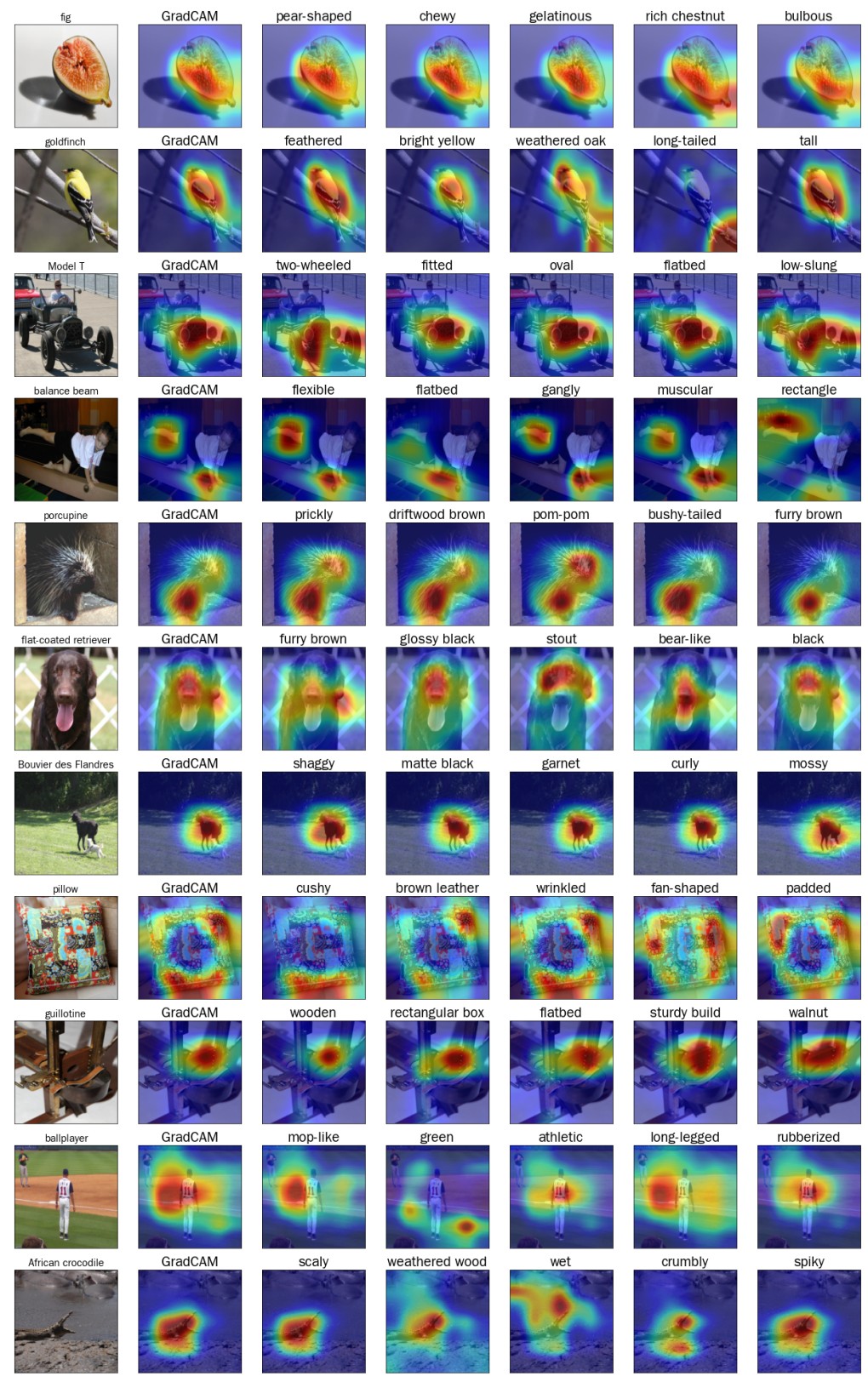

Figure 8: **ImageNet results of TextCAM with saliency groups.** Each column displays, respectively, the original image, GradCAM result, and grouped saliency maps along with their corresponding text explanations from the top-5 TextCAM results.

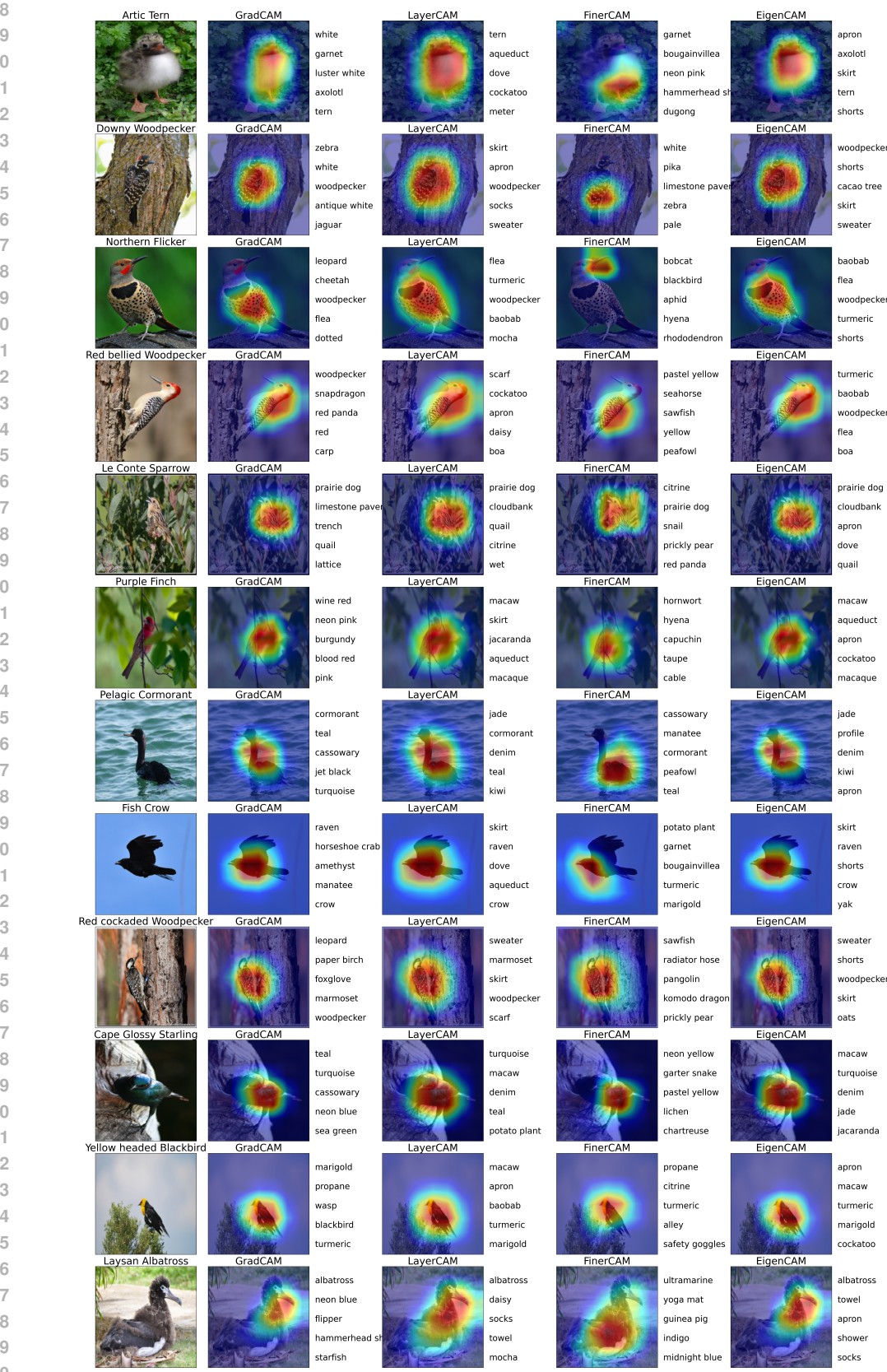

Figure 9: **CUB-200 results of TextCAM**

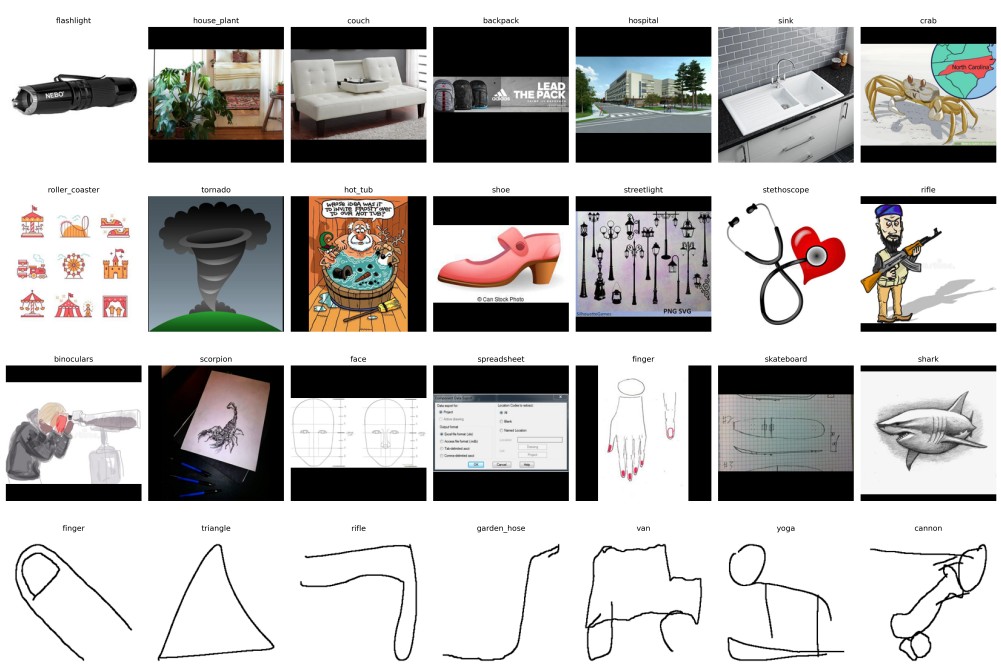

Figure 10: Image examples from DomainNet. From top to bottom are *real, clipart, sketch and quickdraw*.

