# OpenReview forum: "TextCAM: Explaining Class Activation Map with Text"
_ICLR.cc/2026/Conference — ICLR 2026 Conference Withdrawn Submission_

### Official Review · Reviewer_XDQu · 2025-10-24

**Soundness:** 1
**Presentation:** 3
**Contribution:** 1
**Rating:** 2
**Confidence:** 4

**Summary:**

The paper tries to explain the underlying features a neural network looked at using natural language, aimed as an extension to CAM like techniques. The method dissects the dataset based on activation of the specific layer being probed, by projecting the nd layer activation into a single dimension. Using this activation score, top M scoring samples (positive) and least M scoring samples (negative) are selected. CLIP embeddings are computed for those activations, then LDA  finds a vector that divides the positve and negative samples in the CLIP space, which is then weighted with the activation response and then with the weights from CAM like techniques. Then the embedding is matched with a possible natural language concept/explanation, to generate the final natural language explanation. Experiments are performed on Imagenet, CUB-200 and CLEVR datasets.

**Strengths:**

1. The intuition that CAM like saliency maps dont uncover specific features used by the network is valid, and the use of natural language to try to explain these features is valid to an extent.
2. The paper details implementation and design choices aptly.
3. The writing is mostly consistent

**Weaknesses:**

1. Novelty: This is not the first paper that tries to explain the  using natural language, previous works have tried to replace a concept bottleneck layer to produce textual explanations[1], while this paper extends the natural language explanations to intermediate layers, CLIP is designed to represent the vision-language co-space, it may not do well to represent complex intermediate patterns generated by the neural network, therefore, this method is useful usually only in the final layers, which is not novel.

2. "Analyze Per-channel Response Patterns." step in section 3.2 doesnt consider superposition[2].  The technique splits the entire dataset by a score that is basically how much did the layer activate for that particular sample, but a layer doesnt activate for just one feature, therefore just using this activation score on all the classes together would lead to multiple features being present in positive M. Therefore, since the binary positive and negative sets dont represent linearly separable classes in the clip space, LDA is likely to generate instable projection vectors.
3. Like the paper correctly states, these textual features are mostly guesses, and are not reliable. When CAM fails, these explanations fail, moreover, its not certain that when CAM succeeds, the text generated to support the CAM is correct, especially given point 2. The paper tries to ablate this in Section 4.3.1 but the data is mostly toy data. The paper can try to generate subtle modifications to  images using diffusion models and then try counterexample experiments but given point 2, the method is still largely unreliable.
4. As discussed in point 1, this method is mostly useful in examining non complex features, given the simple language in the CLIP space, the uses in final layer is not that helpful because the text is automatically inferred from the image, mostly its doing a kind of zero shot classification in the last layer.

References:
1. Yamaguchi, S., & Nishida, K. (2025). Explanation Bottleneck Models. arXiv [Cs.AI]. Retrieved from http://arxiv.org/abs/2409.17663
2. https://transformer-circuits.pub/2022/toy_model/index.html

**Questions:**

1. How does the paper exactly compute per sample projection ? Or does it just use the dataset wide LDA projection,?

---

### Official Review · Reviewer_uU83 · 2025-10-28

**Soundness:** 3
**Presentation:** 3
**Contribution:** 2
**Rating:** 4
**Confidence:** 4

**Summary:**

The paper introduces TextCAM, an explainability framework that enriches Class Activation Maps (CAMs) with natural language descriptions to improve the interpretability of deep vision models. While traditional CAM-based methods highlight the spatial regions influencing a model’s decision, they fail to clarify what visual attributes drive these activations. TextCAM bridges this semantic gap by combining the spatial localization of CAM with the semantic alignment capabilities of vision–language models (VLMs) like CLIP.

**Strengths:**

Strengths:

- Modular, training-free design: This means that it can be directly applied to existing models with almost no additional cost.

- TextCAM not only provides global interpretation, but also subdivides saliency maps into multiple semantic regions (such as "long neck", "feathers", "legs"), providing finer grained explanatory visualizations.

- It can be used in conjunction with any CAM method (Grad CAM, Layer CAM, Eigen CAM, etc.) and supports CNN and Transformer structures.

**Weaknesses:**

Weakness:

- It seems that using language to assist vision visualization is a very new idea or insight (of course, if the author has evidence, they can list it to convince me).

- The author mentioned that Grad-CAM-like procedures are unstable (**w**) in the Transformer structure, causing fluctuations in TextCAM interpretation on ViT. What is the deeper reason for this? Have you tried using VLLM representation?

- Although TextCAM is training free, in actual deployment, it requires perform LDA (O (channels x feature-dim ² complexity) on each channel. Performing sparse optimization (ADMM solution) on a large vocabulary has a high computational cost and is not suitable for large-scale online interpretation scenarios.

- It seems that the paper did not provide standard CAM explanatory evaluation indicators such as fidelity metrics (e.g., delivery/insertion AUC) or localization accuracy?

**Questions:**

Can you explain the main difference of this work between [1]? For example, the significant innovation or advantages of this article compared to [1]. I don't seem to see any significant innovation on top of it, as you emphasized at the beginning of the article using language to enrich CAM. This is a big confusion for me. There should be a lot of work related to using text/language to assist visualization (I may only remember seeing an article from 2021). Compared to them, your emphasis on using language to enhance innovation seems a bit weak? Also, can you compare with them? Showcasing one's strengths from certain dimensions?

Beyond CLIP, why not try using a VLLM (Vision Large Language Model)? CLIP is just an instance of VLMs. Does your method only compatible with CLIP or is the model-agnostic?


> [1] Generic Attention-model Explainability for Interpreting Bi-Modal and Encoder-Decoder Transformers 2021

---

### Official Review · Reviewer_p88y · 2025-11-01

**Soundness:** 2
**Presentation:** 3
**Contribution:** 3
**Rating:** 4
**Confidence:** 5

**Summary:**

The paper proposed an language argumentation method for CAM(class activation mapping) based explanation models, which is desired in current explainable AI techniques. The basic idea is to make use of the element-wise activation map of CAM as semantic units and adopt CLIPs to generate text explanations based on decomposed images.

**Strengths:**

Although somehow primitive, the idea of using CAM to decompose image elements are interesting and promising for generate rich semantic explanations.

The paper proposed novel methods for optimizing text selection from CLIP results for purpose of explanation.

**Weaknesses:**

Without proper processing, the CAM channels generally does not have clear semantic meaning, which hindered the usefulness of the approach. As we can see from the results, the generated text comprises high portion of errors. It would be beneficial to introduce image segmentation and knowledge bases (e.g., ontology) to improve the results.

There are already some similar works which uses large language models to interpret the image explanation results (e.g., MONET, Nature Medicine 2024, 1154). The authors should try comparing with them.

The survey on xAI methods is limited. Especially, counter-factual generation methods and concept-based explanation methods are not mentioned. Both also fit your framework. It would be potentially very promising to incorporate concept-based method to enhance your approach.

**Questions:**

Why you encode only vocabulary at the first step? This will incur significant loss of semantic information.

---

### Official Review · Reviewer_xjVc · 2025-11-01

**Soundness:** 3
**Presentation:** 2
**Contribution:** 2
**Rating:** 4
**Confidence:** 4

**Summary:**

This paper introduces TextCAM, a method that enhances traditional Class Activation Mapping (CAM) by integrating natural language descriptions. TextCAM first compute channel-level semantic representations via CLIP embeddings and linear discriminant analysis. Then a sparse text selection and grouping strategy is introduced to generate saliency maps with more interpretable text descriptions from text-annotated saliency groups.

**Strengths:**

1. The grouping of saliency maps based on semantic concepts is a useful extension, providing more detailed and organized insights into what the model highlights

2. The paper demonstrates results across multiple datasets

**Weaknesses:**

1. Lack of Technical Depth (Main concerns):

For the technique on "Generate Semantic Representation", it is extended from the GAP and CAM, while relying heavily on pre-trained CLIP embeddings and standard LDA techniques. The contribution often feels incremental rather than groundbreaking;

Sparse optimization for text selection is built upon text-driven visual-language relevance estimation, which is common-used on both CLIP-based and GPT-based methods.

2. Over-Reliance on CLIP:

TextCAM is tied to the quality of CLIP's pre-trained embeddings, which raises concerns about its applicability to tasks or domains where CLIP's semantic space is less effective. The use of CLIP embeddings introduces a dependency, how about GPT-based VLMs?

3. The use of LDA for channel-wise analysis and sparse optimization for text selection introduces computational overhead, lack of analysis on large-scale datasets

4. The vocabulary construction process lacks rigor and seems arbitrary, how can we establish it?

5. What are the failure cases of TextCAM? Are there scenarios where it produces misleading or incoherent explanations?

**Questions:**

1. Strengthen the analysis on technical contributions

2. Test the method on more challenging datasets and tasks, such as multi-object scenes, occlusions, or ambiguous contexts.

3. systematically identify and address scenarios where TextCAM fails to provide meaningful explanations.

4. If possible, include human evaluations to validate the interpretability of the textual explanation (not necessary)

---

### Official Review · Reviewer_71st · 2025-11-01

**Soundness:** 3
**Presentation:** 3
**Contribution:** 2
**Rating:** 4
**Confidence:** 3

**Summary:**

Class activation maps (CAM) provide little insight into what attributes underlie activations. In this work, the authors combine vision-language models with CAM in order to produce textual descriptions of salient visual evidence. Results show that the proposed approach yields accurate descriptions.

**Strengths:**

- This work addresses an important problem - improving explainability of models by generating text descriptions from class activation maps
- The paper is well-written and the methods are clearly explained

**Weaknesses:**

- **Insufficient analysis:** Quantitative results are limited, and whereas results in Section 4.3.1 are compelling, evaluations are performed on a small set of synthetic settings. This is evidenced by the fact that the accuracy of the proposed method is 100%. The paper could have benefitted from more complex quantitative evaluations to precisely highlight the strengths and limitations of the proposed method.
    - While qualitative results in Section 4.2 are interesting, it is difficult to know whether the generated attributes are accurate.
    - Do the trends observed on the CLEVR benchmark in Section 4.3.1 hold across different choices for the CAM method?
    - It appears that all of the evaluations in Section 4 (with the exception of Section 4.4) are performed using a ResNet-50 model. Do trends hold under different choices for the CNN?
- **Method design:** Design choices for the proposed method are not sufficiently evaluated or ablated, making it challenging to determine the utility of each contribution in Section 3.

**Questions:**

Questions are listed above under weaknesses.

---

### Official Review · Reviewer_QHAv · 2025-11-01

**Soundness:** 2
**Presentation:** 2
**Contribution:** 2
**Rating:** 4
**Confidence:** 4

**Summary:**

The TextCAM paper proposes enriching traditional Class Activation Map (CAM) visual explanations with natural-language descriptions derived from vision–language models (VLMs) such as CLIP. By mapping channel-level activations to semantic embeddings using Linear Discriminant Analysis (LDA), TextCAM generates textual explanations corresponding to salient image regions. The paper also introduces a grouping strategy to cluster channels into semantically coherent saliency groups, offering fine-grained, text-annotated visual explanations.

**Strengths:**

The paper addresses an important gap in CAM-based interpretability by adding semantic meaning to purely visual heatmaps.

The method is training-free and can be combined with any existing CAM variant, making it broadly applicable.

Integration of CLIP embeddings is a reasonable and practical way to connect feature maps with human-understandable attributes.

**Weaknesses:**

The key idea, mapping latent features or channels to textual concepts using a joint vision–language embedding space, has already been explored in earlier works such as LaViSE (CVPR 2022) and other filter-level semantic alignment methods. TextCAM’s contribution lies primarily in applying this concept to CAM-based spatial localization, rather than developing a new alignment or representation mechanism. LaViSE also provided unsupervised textual descriptions for filters and bias analysis applications, which conceptually overlap with TextCAM’s goal. The omission of this related work makes the contribution appear less grounded and potentially redundant.

The combination of CAM weights with CLIP-based LDA projections is empirically motivated but lacks formal justification. The sparse optimization and grouping procedures are heuristic and not well-analyzed in terms of stability, interpretability, or robustness.

The experiments primarily show qualitative visualizations. Claims of improved interpretability and model faithfulness are not supported by objective metrics (e.g., human studies, consistency scores, or statistical tests).

The figures mostly present clear or intuitive cases where the text matches the highlighted region, but do not include examples where the model fails or produces nonsensical descriptions. The authors also need to provide some error cases.

**Questions:**

Could you clarify the conceptual or methodological innovation that distinguishes your approach beyond combining CAM localization with CLIP-based textual projection?

The paper employs Linear Discriminant Analysis (LDA) to project channel activations into the CLIP text space. Why was LDA chosen over other semantic alignment strategies (e.g., attention-based, contrastive, or regression-based mappings)? How sensitive are the results to the number of LDA components or to the choice of semantic embedding space?

The paper presents visually convincing examples, but how does TextCAM behave when the CAM highlights ambiguous or overlapping regions, or when the model misclassifies an image? Can you provide qualitative or quantitative error analysis to show robustness in such cases?

---

### Note · Authors · 2025-11-12

I have read and agree with the venue's withdrawal policy on behalf of myself and my co-authors.